# Occurrence of Methicillin-Resistant Coagulase-Negative Staphylococci (MRCoNS) and Methicillin-Resistant *Staphylococcus aureus* (MRSA) from Pigs and Farm Environment in Northwestern Italy

**DOI:** 10.3390/antibiotics10060676

**Published:** 2021-06-05

**Authors:** Miryam Bonvegna, Elena Grego, Bruno Sona, Maria Cristina Stella, Patrizia Nebbia, Alessandro Mannelli, Laura Tomassone

**Affiliations:** 1Department of Veterinary Sciences, University of Turin, Largo P. Braccini 2, 10095 Grugliasco, Italy; elena.grego@unito.it (E.G.); mariacristina.stella@unito.it (M.C.S.); patrizia.nebbia@unito.it (P.N.); alessandro.mannelli@unito.it (A.M.); laura.tomassone@unito.it (L.T.); 2Local Veterinary Service, Animal Health, ASL CN1, Via Torino, 137, 12038 Savigliano, Italy; bruchi.berso@gmail.com

**Keywords:** MRCoNS, MRSA, *mecA*, PBP2-a, swine farming, antimicrobial resistance

## Abstract

Swine farming as a source of methicillin-resistant *Staphylococcus aureus* (MRSA) has been well documented. Methicillin-resistant coagulase-negative staphylococci (MRCoNS) have been less studied, but their importance as pathogens is increasing. MRCoNS are indeed considered relevant nosocomial pathogens; identifying putative sources of MRCoNS is thus gaining importance to prevent human health hazards. In the present study, we investigated MRSA and MRCoNS in animals and environment in five pigsties in a high farm-density area of northwestern Italy. Farms were three intensive, one intensive with antibiotic-free finishing, and one organic. We tested nasal swabs from 195 animals and 26 environmental samples from three production phases: post-weaning, finishing and female breeders. Phenotypic tests, including MALDI-TOF MS, were used for the identification of *Staphylococcus* species; PCR and nucleotide sequencing confirmed resistance and bacterial species. MRCoNS were recovered in 64.5% of nasal swabs, in all farms and animal categories, while MRSA was detected only in one post-weaning sample in one farm. The lowest prevalence of MRCoNS was detected in pigs from the organic farm and in the finishing of the antibiotic-free farm. MRCoNS were mainly *Staphylococcus sciuri*, but we also recovered *S. pasteuri*, *S. haemolyticus*, *S. cohnii*, *S. equorum* and *S. xylosus.* Fifteen environmental samples were positive for MRCoNS, which were mainly *S. sciuri*; no MRSA was found in the farms’ environment. The analyses of the *mecA* gene and the PBP2-a protein highlighted the same *mecA* fragment in strains of *S. aureus*, *S. sciuri* and *S. haemolyticus.* Our results show the emergence of MRCoNS carrying the *mecA* gene in swine farms. Moreover, they suggest that this gene might be horizontally transferred from MRCoNS to bacterial species more relevant for human health, such as *S. aureus*.

## 1. Introduction

Antimicrobial resistance (AMR) is a global concern to human, animals, and environmental health [1]. Livestock can be reservoir of different antibiotic-resistant bacteria. Methicillin-resistant *Staphylococcus aureus* (MRSA) has been well documented in pigs since its first detection in 2005 in the Netherlands [2] and can be found in the intensive swine farm system across Europe, USA, and Asia [3,4,5,6]. The high MRSA colonization rate among industrially raised pigs poses a threat for farm workers and to people living in high farm-density areas [7,8,9,10].

Coagulase-negative staphylococci (CoNS) are generally considered as commensal non-pathogenic bacteria, belonging to the skin microbiota of different animals [11,12]. However, they have been recently documented as source of severe conditions, such as exudative epidermitis in piglets [13] and mastitis, endocarditis and osteomyelitis in other farm animals [14,15,16]. Furthermore, they are gradually becoming more relevant from a clinical point of view, as cause of hospital-related human infections [17]. Like the coagulase-positive staphylococci (CoPS), CoNS can have a methicillin-resistant phenotype, often due to the presence of the *mecA* gene. This gene is responsible for the expression of a modified penicillin-binding protein (PBP2-a), which has a low affinity for β-lactam antibiotics, like cephalosporins [18]. Indeed, *mecA* gene is responsible for the most clinically relevant antibiotic resistance mechanism in *S. aureus.* Analysis of the evolution of this gene showed that it probably originated from the native *mecA1* present in *S. sciuri* group, which underwent recombination and mutation events [19,20]. The *mecA* gene is located on a mobile genetic element: the staphylococcal cassette chromosome *mec* (*SCCmec*) in *Staphylococcus* spp., and on a *SCCmec*-like element in other species, such as *Macrococcus caseolyticus* [21].

Methicillin-resistant CoNS (MRCoNS) in livestock were firstly reported in healthy chickens in Japan in 1996 [22]. Since then, few investigations have been focusing on these bacterial species in farm animals. For example, methicillin-resistant *Staphylococcus sciuri* were reported in 6.5% of pigs in 10 Belgian farms [23], and ten different MRCoNS species were isolated, along with *S. aureus*, from nasal swabs and farm dust samples in swine farms in The Netherlands [24].

In Italy, scarce information is available on MRCoNS in livestock, and no study has elucidated their presence in swine farms. The initial aim of our study was to unravel the presence of MRSA in pigs at different production stages and in the farm environment, in an area of intensive pig farming in northwestern Italy (Piedmont region, Cuneo province). Due to the massive presence of MRCoNS, we focused on both bacterial groups, sequencing the *mecA* gene to find similar nucleotide mutations in the different staphylococcal strains that can support horizontal gene transfer events. Biosecurity and general farm management were analysed to evaluate possible impacts on the occurrence of MRCoNS and MRSA.

## 2. Results

### 2.1. Biosecurity and Management of Farms

From the analysis of the questionnaires, we appreciated a homogeneous level of general biosecurity in the five visited farms. All the farms claimed an external animal remount from only one gilts’ supplier, with the exception of farm G that had an internal remount. All farmers used to isolate new animals in quarantine. Looking to animal management through the production cycle, it emerged that all farmers used to mix different animal groups through the production cycle, especially piglets from different litters. Only the owner of farm T declared to mix diverse animal groups during the finishing stage. Considering the farm hygiene, all the farmers claimed to adopt a cleaning protocol, however nobody used dedicated clothes and boots to enter the different animal sectors, apart the workers from farm B. Regarding the storage of carcasses, the dedicated refrigerated room was near the animal sectors in all farms, with the exception of farm T. The animals were kept on slatted or partially slatted floor, with the exception of farm S, where pigs were on straw bedding with a minimum slatted part.

### 2.2. Laboratory Analyses

Overall, 127 MRS (mannitol-fermenting on MSA) were recovered from 195 swine nasal swabs (65.1%, 95% CI: 58.0–71.8; Table 1). MRS were recovered in all farms. MALDI-TOF MS bacterial species identification was confirmed by 16S rDNA sequencing; only one post-weaning environmental sample of farm B, initially considered *S. xylosus*, was identified as *S. cohnii* after sequencing.

*Staphylococcus aureus* was isolated from one nasal swab, in the post-weaning phase of the intensive farm T (Table 1). The other MRS isolated were MRCoNS, of which 88.2% (95% CI: 82.6–93.8) were identified as *S. sciuri*. *Staphylococcus pasteuri* (3.9%, 95% CI: 0.5–7.3), *S. haemolyticus* (3.1%, 95% CI: 0.1–6.2), *S. cohnii* (1.6%, 95% CI: 0.0–3.7), *S. equorum* (1.6%, 95% CI: 0.0–3.7), and *S. xylosus* (0.8%, 95% CI: 0.0–2.3) were isolated from nasal swabs as well. The prevalence of MRS was significantly different among farms (Fisher’s Exact test, *p* < 0.001). Indeed, they were particularly abundant in farm T, where 44 out of 45 nasal swabs were positive and one isolate was identified as MRSA. The lowest number of MRS isolates was recovered from the organic farm S (8/15) and at farm P (17/45) (Table 1). MRS prevalence also significantly differed among productive stages (*p* < 0.001). The majority of positive samples was collected from the sows, followed by the post-weaning phase; finishing animals had a lower MRS prevalence, with the lowest number of positives (2/15 swabs) in the antibiotic-free finishing of farm G (Table 2).

Fifteen out of 26 samples collected from the farms’ environment were positive for MRCoNS (Table 1). MRS were recovered in the environment of the farms, except from the organic farm S and the antibiotic-free finishing stage of farm G (Table 2). Again, *S. sciuri* was the main species isolated (*n* = 12), and it was the unique species recovered in the environment in three out of the four positive farms. In farm B, we also identified *S. cohnii*, *S. haemolyticus* and *S. equorum*. In accordance with the animal swabs’ results, MRS were mainly identified in the sows and post-weaning environment. 

PCR amplicons were confirmed as the *mecA* type of *mec* gene through sequencing. 

The analysis of the 527 bp *mecA* amplicons, obtained from 59 selected strains, revealed that all *mecA* sequences were highly related to the reference *S. aureus mecA* sequences (strain N315, COL and MW2). The percentage of identity to the reference strains was higher than 99% in all the tested sequences. The nucleotide alignment revealed four point-mutations in the non-penicillin binding domain (non-PBD) (see Appendix A). 

The most frequently detected mutation was at G737A (missense), which was also present in the reference strain COL (CP000046.1). This point mutation was present in all five farms across fifty-one strains of diverse staphylococcal species (*S. cohnii*, *S. equorum*, *S. haemolyticus*, *S. pasteuri*, *S. sciuri* and *S. xylosus*), in all productive steps, and was also recovered from environmental bacterial strains. In farm P and S, we detected only this mutation. Moreover, the nucleotide mutation T675A was detected in farm B, in two post-weaning *S. cohnii* strains (B1PAS15 and B1PHS1) and in two *S. haemolyticus* samples from the environment and from a finishing pig (B1FHS1 and B1FAS15). *S. equorum* strains (two from animals: B1SAS3, B1SAS15; one from environment: B1SHS1) from farm B had two nucleotide variations: T667G and G737A; this double mutation was detected in the reference *S. equorum* strain SMK37o (GU301099.1). Finally, in farm T we detected two point mutations, T675A and G682A, in four identical *mecA* fragments from animals’ samples, namely a *S. aureus* (T1PAS3), a *S. sciuri* (T1PAS4) and two *S. haemolyticus* (T1FAS7 and T1FAS15). 

The analysis of the aminoacidic sequences revealed the presence of four mutations: Y223D, S225R, A228T and G246E. The mutations Y223D and G246E were detected together only in *S. equorum* strains (B1SAS3, B1SAS15 and B1SHS1), such as is in the reference *S. equorum* SMK37o sequence (ADB44836.1). The mutations S225R, A228T were detected in four strains from farm T, one *S. aureus* (T1PAS3) and one *S. sciuri* from post-weaning (T1PAS4) and two *S. haemolyticus* from finishing (T1FAS7 and T1FAS15). In farm B, four bacterial strains presented the aminoacidic substitution S225R: two *S. haemolyticus* from finishing (B1FHS1 and B1FAS15) and two *S. cohnii* from post-weaning (B1PAS15 and B1PHS1) (Table 3). The mutation G246E was the most frequently detected in bacterial strains from all five farms and all the productive stages in animals and environment (86.4%, 95% CI: 75.0–94.0). S225R was detected in 13.6% (95% CI: 6.0–25.0) of the 59 selected strains, while A228T in 6.8% (95% CI: 1.9–16.5) and Y223D in 5.1% (95% CI: 1.1–14.1) (Table 3).

## 3. Discussion

The objectives of our study were to investigate the presence of MRSA and MRCoNS in pigs at different production stages and in their farm environment, in an area of intensive pig farming of Italy (Piedmont region, Cuneo province). The detection of only one MRSA positive sample out of almost 200 tested animals was unexpected. Indeed, the prevalence of MRSA in finishing pigs in another region of northern Italy, Lombardy, was recently estimated at 17.5% [25]. Furthermore, in the south of Italy, prevalences higher than 45% were reported, with the majority of positive samples from intensively reared animals [26,27]. 

Although our limited sample size may have led us to underestimate the prevalence, our results indicate a rare presence of MRSA in pig farms in our study area. On the other side, we highlighted the massive presence of MRCoNS in pigs and in their environment, especially in the non-antibiotic-free farms. MRCoNS prevalence in intensive and organic farms’ animals was 64.6% overall, much higher than in previous studies from other European countries, where prevalence varied from 36.3% in Switzerland to 6.5% in Belgium [11,23]. The breeding stage (sows) showed the highest MRS prevalence (80%, Table 2). This may be explained by the older age of these animals compared to the other productive categories, so that they are possibly subjected for a longer time to antibiotic treatments. During their lifetime, sows can indeed manifest different clinical problems at the respiratory and reproductive system and suffer from joint diseases, requiring antibiotic treatments [28]. Post-weaning was the second most colonised phase by MRS, with 71.7% of positive samples. Colonization in this stage could be determined by different reasons: 1. the young age of the animals, which are probably more susceptible to MRS due to their immature nasal microflora [29]; 2. the fact that piglets from different sows are mixed after the farrowing stage and can exchange bacteria [29,30,31]; 3. post-weaning stress; 4. environmental contamination [29]; 5. antimicrobial treatment [30,31]. All the farmers stated that animal mixing, especially piglets, was a common practice, and this could be considered a risk factor for MRS spread, especially in the post-weaning phase. Another contributing factor could be the use of the same clothes and boots to visit the different productive phases in the farm. This habit was common in almost all our farms and could contribute to the dissemination of bacteria across the farm sectors. For example, in farm P, *S. sciuri* with the *mecA* gene G246E mutation was detected in all three animal sectors, while *S. pasteuri* harbouring the same mutation was sampled on finishers and in one weaned animal in farm T. The general use of slatted floor among farms was another potential risk factor for the dissemination of MRS among animals, as was elucidated in previous studies regarding MRSA [32].

In our research, *S. sciuri* was the predominant species among MRCoNS in animals and in the farm environment. This is in agreement with other studies on animals, sewage and dust in swine farms in Europe and Asia [33,34]. We hypothesize that the predominant presence of MR-*S. sciuri* in the farm environment is linked to *S. sciuri* nasal colonisation of the animals, due to their natural nuzzling behaviour. Indeed, this bacterium is a well-fitted free-living microorganism, that can be found in a wide range of hosts [35]. 

The massive presence of MR-*S. sciuri* on the nasal mucosa might negatively affect MRSA colonisation. Indeed, a natural inhibition of *S. aureus* in humans with a previous nasal colonisation by a commensal CoNS, *Staphylococcus epidermidis*, was demonstrated [36]. However, additional studies are needed to understand if the presence of MRCoNS may inhibit a successful nasal colonisation by MRSA even in pigs. This natural inhibition could explain the extremely rare presence of MRSA among our study animals, together with management practices in the farms, such as the animal remount system. In fact, previous studies have documented that using multiple animal suppliers is a risk factor for MRSA colonization in the farm animals [37]. All our farmers instead declared to have only one gilts’ supplier, or to have an internal remount (Farm G), and this could positively influence the animal negative MRSA status. 

Contrary to the high prevalence in animals’ samples, MRCoNS prevalence in the farm environment, in all the production phases, was in line with other European studies, where a prevalence up to 64% was found (e.g., in Germany [33]). 

Finishing environment, like finishing animals, was the less contaminated with MRS, with only three bacterial strains isolated from a total of 10 samples (Table 2). This can be explained by the infrequent antibiotic treatments during this productive stage, due to the approaching slaughtering. 

Although we did not recover the simultaneous presence of MRCoNS and MRSA in the same animal, we found them in the same productive stage of one farm (farm T). This can be considered a risk for the horizontal gene transfer of *mecA* from one staphylococcal species to *S. aureus*, that is a well-established human pathogen. The mobilization of *SCCmec* cassette, including the *mecA* gene, between CoNS and *S. aureus* was demonstrated by previous studies; the same nucleotide sequence was detected in various staphylococcal strains and species, indicating that this genetic element can move among staphylococci [38,39,40,41,42]. In vivo *mecA* mobilization from a CoNS species to *S. aureus* was demonstrated in a neonate, with the detection of the same *mecA* restriction patterns in a MRSA and a MR- *S. epidermidis* isolate [43]. 

In farm T, the same mutated *mecA* fragment was recovered from different staphylococcal species from two diverse productive phases: MRSA (T1PAS3) and MR-*S. sciuri* (T1PAS4) from weaned animals and two *S. haemolyticus* strains from finishers (T1FAS7 and T1FAS15). To our knowledge, this mutation had never been reported in *S. haemolyticus*; it was only previously detected in pigs in *S. aureus* strains from Denmark (CP028163.1 and CP028190.1) and China (CP065194.1). Moreover, the mutated *mecA* fragments of all *S. pasteuri* strains (T1FAS1, T1FAS4, T1FAS6 and T1FAS13) in this same farm, had never been reported from swine.

Farm B displayed the highest diversity of MRS species in animals and in the environment. The mutated *mecA* gene found in *S. equorum* strains (B1SAS3, B1SAS15 and B1SHS1) was identical to the one recovered from a cat in the Netherlands in 2005 (GU301099.1). 

The *mecA* genes sequenced in *S. haemolyticus* (B1FHS1 and B1FAS15) and *S. cohnii* (B1PHS1 and B1PAS15) strains were identical to two MR-*S. haemolyticus* isolated in China from pathological bovine milk (KM369884.1 and KM369884.1) and from a swine nasal swab sampled in China (CP063443.1). The same *mecA* sequence was found in MR- *S. haemolyticus* isolated from human blood (AB437289.1) and urine (CP052055.1) samples. From our knowledge, this *mecA* nucleotide sequence is here first reported in *S. cohnii*.

The identified nucleotide mutations led to aminoacidic substitutions that had been previously recovered from human clinical specimens and had been correlated, with other variations in the non-PBD of the PBP2-a protein, to resistance to fifth generation cephalosporins [44,45,46]. The highly recurrent G246E mutation was also the most frequently reported in a study from Algeria, where it was recovered in *S. aureus, S. sciuri, S. saprophyticus* and *S. lentus* strains collected from human nasal samples [47]. Considering the available scientific literature, we here first report this mutation in the PBP2-a from *S. xylosus* (T1SAS12) and from *S. cohnii* (P1SAS14) of animal origin; this variation in *S. cohnii* has been hitherto detected from human clinical samples (ADM43473.1).

The S225R mutation, that we highlighted in *S. cohnii* and *S. sciuri* strains, had been previously recovered in *S. haemolyticus* and *S. aureus* strains [45].

In conclusion, our study highlights an unexpected number of mutations in the *mecA* gene from swine MRS, some of which had never been detected in staphylococcal species from pigs. Consequently, monitoring of MRS at farm level is relevant to understand the risk for farmers to acquire these bacteria. The possibility for swine farmers to be colonised with MRCoNS, due to occupational exposure, was documented in a recent study [48]. The finding of the same MRCoNS in the animals and environmental samples suggests that the farm environment can be a source of animal contamination, or that animals can contaminate the environment, due to their massive colonisation with these bacteria [24]. Indeed, in Italy, MRSA colonised pigs seem to be the principal vehicle of transmission of MRSA to the environment [49]. 

Although we sampled a small number of farms, our results prove a significant colonization of pigs with MRS in different productive stages in the study area, where our farms represent the standard production typology. We will further investigate whether the antibiotic usage in the farms can be related to the MRS prevalence detected. Indeed, the misuse of antibiotics, in particular extended-spectrum cephalosporins and aminopenicillins, can contribute to the selection of methicillin-resistant bacteria in swine farms [50,51]. 

The unpredicted massive detection of MRCoNS in this area of Italy in pigs and their farm environment, with the contemporary presence of MRSA and MRCoNS, underline the need of monitoring both bacterial groups, since they can possibly transfer the *mecA* gene between them and can colonise human hosts. Furthermore, the finding of the same *mecA* genes in *S. aureus* and other swine-related species such as *S. sciuri*, support the role of these last bacteria as reservoir of *mecA* gene. Further studies are necessary to understand the possibility of horizontal gene transfer among staphylococci at farm level and the possible negative effect of MRCoNS on MRSA nasal colonization in pigs.

## 4. Materials and Methods

### 4.1. Farm Samples’ Collection

The study was carried out in an industrial farming area in Cuneo province, Piedmont region [52]. Farms were representative of the standard pig farms present in the area, according to the veterinarians of the Local Veterinary Health Service (ASL CN1). Farms were chosen based on a convenience sampling, considering the willingness of the farmer to collaborate, the production type (intensive, organic and antibiotic-free) and cycle (close, farrow-to-finish; or open, finishers only). We selected five farms: three intensive close (farrow-to-finish) farms (named B, P, T), one intensive close farm, antibiotic-free at finishing (G), and one organic finishers-only farm (S). Samples were collected between October 2019 and September 2020, in occasion of routine veterinary checks. The sample size was calculated to detect at least one MRSA positive sample per farm, based on a minimum expected MRSA prevalence of 10% and considering a 95% confidence level. During each sampling, we collected 15 nasal swabs from each animal category present in the farm (post-weaning, finishing and sows); within each pen, the sampled pigs were randomly chosen. Moreover, we sampled two environmental swabs for each productive stage, from sites in tight contact with animals like bed pavements, troughs, and barriers, and on places around the animals like the pigsty walls, floor corners and tubes. Each sample was identified with a code indicating the farm name (B,G,P,S and T), number of sampling (1), productive phase (F = finishing, P = post-weaning and S = sows), source (A = animal and H = environment), bacterial genus (S = *Staphylococcus*) and a progressive number (1–15 for animals, 1–2 for environment). In the organic farm S, only 15 animals and 2 environmental samples were taken, since it was a finishers-only farm. Samples were kept in a refrigerated box till the arrival in the laboratory and were processed within 24 h.

Prevalence of positive samples was calculated, with 95% confidence intervals (95% CI). Fisher’s Exact test was used to assess differences in MRS prevalence among different farms and productive stages. Statistical analyses were performed with R software (R Core Team, 2020 [53]).

### 4.2. Biosecurity and Management Data Collection

A questionnaire about general farm biosecurity was compiled when farms were visited to evaluate animal flow in the different farm sectors, remount, piglets mixing from different litters, gilts’ quarantine, use of dedicated clothes and boots to enter the different animals’ sectors, cleaning protocol, floor type and the carcasses management (see Appendix A). 

### 4.3. Phenotypic Analysis

Each swab was subjected to an enrichment stage in a liquid medium. Tryptic soy broth (TSB) (Oxoid, Wade Road Basingstoke, UK) with 2.5% of NaCl [54] was used with the addition of two antibiotics: cefoxitin (3.5 mg/L) (Sigma-Aldrich, St. Louis, MO, USA) and aztreonam (20 mg/L) (Sigma-Aldrich, St. Louis, MO, USA). Cefoxitin was added to select MRS, while aztreonam to inhibit Gram-negative bacteria growth. Swabs were immersed for 5 min in 4 mL of this broth; afterwards, broth samples were placed in a shaker incubator for 24 h at 35–37 °C at 220 rpm. After the enrichment step, a loop of 10 μL of the liquid samples were spread on a selective solid medium; the medium was Mannitol Salt agar (MSA) with 6% NaCl ([55]), plus cefoxitin (3.5 mg/L) (Sigma-Aldrich, St. Louis, MO, USA). MSA was prepared with phenol red, 0.025 g/L (Sigma-Aldrich, St. Louis, MO, USA), bacteriological peptone 10 g/L (Oxoid, Wade Road Basingstoke, UK), mannitol, 10 g/L (Sigma-Aldrich, St. Louis, MO, USA) NaCl, 60 g/L, cefoxitin 3.5 mg/L (Sigma-Aldrich, St. Louis, MO, USA) beef extract powder 1 g/L, and agar 15 g/L (Oxoid, Wade Road Basingstoke, UK) with a final pH of 7.4 +/− 0.2. Catalase test, gram staining and oxidase test were used as supportive tests in staphylococcal identification on presumptive round and yellow *S. aureus* colonies that presented mannitol fermentation on MSA. Bacterial strains were stored in 500 µL of TSB plus 15% glycerol at −80 °C.

Phenotypic bacterial identification was performed using matrix-assisted laser desorption and ionization time-of-flight mass spectrometry (MALDI-TOF MS) Microflex™ LRF (Bruker Daltonik GmbH, Bremen, Germany). A modified direct transfer-formic acid method was used for sample preparation as described previously [56]. Briefly, one colony from a fresh pure culture was taken with a disposable loop and spread on a single well of the microplate reader, to have a thin layer. Then, 0.9 μL of acid formic (diluted at 70%) was added on the well. After acid formic was dried, 1 μL of the saturated α-cyano-4-hydroxy-cinnamic acid (HCCA) matrix (Bruker Daltonik GmbH, Bremen, Germany) was added over the well. Finally, after HCCA matrix was left to dry, bacterial samples on the microplate were analysed with MALDI-TOF MS within 24 h. MALDI Biotyper^®^ (Bruker Daltonik GmbH, Bremen, Germany) software was run to classify bacteria at genus and species level. Following Bruker recommendations, specimens with a similarity log-score threshold between >1.7 and <1.999 were classified for presumptive genus, while a score > 2 and < 2.299 were used for secure genus identification, and probable species identification. Results < 1.7 were considered not reliable for bacterial genus identification. 

### 4.4. Genotypic Analysis

DNA was extracted from bacterial colonies using a modified boiling method: briefly, one or two colonies were picked with a sterile loop and immersed in 1 mL of PBS in a 1.5 mL Eppendorf tube; then, samples were centrifuged for 5 min at 13,500 rpm. Supernatant was discarded and the remnant bacterial pellet was mixed with 100 μL of sterile deionized water and vortexed for some seconds. Afterwards, samples were placed in thermoblock for 8 min at 95 °C, and then, stored at −20 °C [57]. Quantity of extracted DNA was measured with a spectrophotometer NanoDrop™ 2000 (Thermo Scientific, Waltham, WA, USA). 

Polymerase chain reaction (PCR) was used to confirm phenotypic methicillin resistance (*mecA* gene) and to verify bacterial identification (16S rDNA gene). To confirm MRSA identity, we used a multiplex PCR protocol, targeting the *mecA* and 16S rDNA genes and the *S. aureus*-specific *nuc* gene [58]. Simplex protocols were used to amplify the *mecA* (527 bp) and 16S rDNA genes (500 bp) for nucleotide sequencing. The 16S rDNA gene was tested to confirm genus in samples with a MALDI-TOF MS log-score between 1.7 and 2. Positive controls (from Turin University Culture Collections) and negative controls (deionised DNA-free water) were added to every PCR reaction.

Amplified fragments of a group of 59 strains, randomly chosen from all the farms and productive stages, were purified with ExoSAP-IT™ PCR Product Clean-up Kit (GE Healthcare Limited, Chalfont, UK) and sequenced in an external laboratory (BMR Genomics, Padua, Italy). Obtained nucleotide sequences were analysed using BioEdit 7.2.5 Sequence Alignment Editor^©^software; multiple alignment with reference sequences was carried out with ClustalW tool. The same software was used to convert nucleotide sequences in aminoacidic sequences. To compare sequences with available sequences in GenBank, we used BLAST^®^ (https://blast.ncbi.nlm.nih.gov/Blast.cgi, accessed on 18 March 2021). Reference strains used in the nucleotide alignment were the MRSA N315 (BA000018.3), MW2 (NC003923.1), COL (CP000046.1) and the methicillin- resistant *S. equorum* SMK37o (GU301099.1).

We deposited *mecA* sequences in GenBank with the accession numbers: MW732662, from MW768093 to MW768105, and MW774905. 

## Figures and Tables

**Table 1 antibiotics-10-00676-t001:** Methicillin-resistant staphylococci (MRS) isolated from swine nasal swabs and the environment in five swine farms in northern Italy, 2019–2020.

Farm ID	Farm Type		MRS *n* Positive Samples /*n* Tested (%; 95% CI)	*Staphylococcus* Species (*n* Positive)						
				*S. aureus*	*S. cohnii*	*S. equorum*	*S. haemolyticus*	*S. pasteuri*	*S. sciuri*	*S. xylosus*
Farm B	intensive	animals	27/45	0	1	2	2	0	22	0
			(60%; 44.3–74.3)							
		environment	4/6	0	1	1	1	0	1	0
			(66.7%; 22.3–95.7)							
Farm G	intensive (antibiotic-free finishing)	animals	31/45	0	0	0	0	0	31	0
			(68.9%; 53.3–81.8)							
		environment	4/6	0	0	0	0	0	4	0
			(66.7%; 22.3–95.7)							
Farm P	intensive	animals	17/45	0	1	0	0	0	16	0
			(37.8%; 23.8–53.5)							
		environment	2/6	0	0	0	0	0	2	0
			(33.3%; 0.4–77.7)							
Farm S	organic	animals	8/15	0	0	0	0	0	8	0
			(53.3%; 26.6–78.7)							
		environment	0/2	0	0	0	0	0	0	0
			(0%; 0–84.2)							
Farm T	intensive	animals	44/45	1	0	0	2	5	35	1
			(97.8%; 88.2–100)							
		environment	5/6	0	0	0	0	0	5	0
			(83.3%; 35.9–99.6)							
Total		animals	127/195	1	2	2	4	5	112	1
			(65.1%; 58.0–71.8)							
		environment	15/26	0	1	1	1	0	12	0
			(57.7%; 36.9–76.7)							

**Table 2 antibiotics-10-00676-t002:** Methicillin-resistant staphylococci (MRS) isolated from animals (nasal swabs) and the environment in five farms in northern Italy, 2019–2020, by productive stage. All isolates were MRCoNS except from one MRSA in farm T (indicated by an asterisk).

Farm ID	Farm Type	MRS Per Productive Stage					
		*n* Positive Samples/*n* Tested (%; 95% CI)					
		Finishing		Post-Weaning		Sows	
		Animals	Environment	Animals	Environment	Animals	Environment
Farm B	Intensive	8/15	1/2	8/15	2/2	11/15	1/2
		(53.3%; 26.6–78.7)	(50.0%; 12.6–98.7)	(53.3%; 26.6–78.7)	(100%; 15.8–100)	(73.3%; 44.9–92.2)	(50.0%; 12.6–98.7)
Farm G	Intensive (antibiotic-free finishing)	2/15	0/2	14/15	2/2	15/15	2/2
		(13.3%; 1.6–40.5)	(0%; 0–84.2)	(93.3%; 68.1–99.8)	(100%; 15.8–100)	(100%; 78.2–100)	(100%; 15.8–100)
Farm P	Intensive	5/15	1/2	6/15	0/2	7/15	1/2
		(33.3%; 11.8–61.6)	(50.0%; 12.6–98.7)	(40.0%; 16.3–67.7)	(0%; 0–84.2)	(46.7%; 21.3–73.4)	(50.0%; 12.6–98.7)
Farm S	Organic	8/15	0/2	-	-	-	-
		(53.3%; 26.6–78.7)	(0%; 0–84.2)				
Farm T	Intensive	14/15	1/2	15/15 *	2/2	15/15	2/2
		(93.3%; 68.1–99.8)	(50.0%; 12.6–98.7)	(100%; 78.2–100)	(100%; 15.8–100)	(100%; 78.2–100)	(100%; 15.8–100)
Total		37/75	3/10	43/60	6/8	48/60	6/8
		(49.3%; 38–60.6)	(30%; 6.7–65.2)	(71.7%; 58.6–82.5)	(75%; 34.9–96.8)	(80%; 67.7–89.2)	(75%; 34.9–96.8)

**Table 3 antibiotics-10-00676-t003:** Point mutations and PBP2-a aminoacidic substitutions in MRS isolated in five farms in northern Italy, 2019–2020, by productive stage (sows, post-weaning and finishing). Reference strains MRSA COL, MW2, N315 and MR-*S. equorum* SMK37o were used.

Strain	Organism	Farm	Productive Phase	Sample	Point Mutation				PBP2-a Mutation			
					T667G	T675A	G682A	G737A	Y223D	S225R	A228T	G246E
B1SAS3	*S. equorum*	B	sows	animal	X			X	X			X
	(MW768099)											
B1SAS5,9,11	*S. sciuri*	B	sows	animal				X				X
B1SAS15	*S. equorum*	B	sows	animal	X			X	X			X
	(MW768100)											
B1SHS1	*S. equorum* (MW768101)	B	sows	environment	X			X	X			X
B1PAS10	*S. haemolyticus*	B	post-weaning	animal				X				X
B1PAS15	*S. cohnii*	B	post-weaning	animal		X				X		
	(MW768093)											
B1PHS1	*S. cohnii*	B	post-weaning	environment		X				X		
	(MW768094)											
B1PHS2	*S. sciuri*	B	post-weaning	environment				X				X
B1FAS9	*S. sciuri*	B	finishing	animal				X				X
B1FAS13	*S. haemolyticus*	B	finishing	animal				X				X
B1FAS15	*S. haemolyticus*	B	finishing	animal		X				X		
	(MW768103)											
B1FHS1	*S. haemolyticus* (MW768102)	B	finishing	environment		X				X		
G1SAS7,12	*S. sciuri*	G	sows	animal				X				X
G1PAS6,15	*S. sciuri*	G	post-weaning	animal				X				X
G1PHS2	*S. sciuri*	G	post-weaning	environment				X				X
P1SAS1,3	*S. sciuri*	P	sows	animal				X				X
P1SAS14	*S. cohnii*	P	sows	animal				X				X
	(MW774905)											
P1SHS1	*S. sciuri*	P	sows	environment				X				X
P1PAS6,12,13,15	*S. sciuri*	P	post-weaning	animal				X				X
P1FAS2,3,9	*S. sciuri*	P	finishing	animal				X				X
P1FHS1	*S. sciuri*	P	finishing	environment				X				X
S1FAS2,7,10,14	*S. sciuri*	S	finishing	animal				X				X
T1SAS2,4,7,10,14	*S. sciuri*	T	sows	animal				X				X
T1SAS12	*S. xylosus*	T	sows	animal				X				X
	(MW768096)											
T1SHS1,2	*S. sciuri*	T	sows	environment				X				X
T1PAS3	*S. aureus*	T	post-weaning	animal		X	X			X	X	
	(MW768098)											
T1PAS4	*S. sciuri*	T	post-weaning	animal		X	X			X	X	
	(MW732662)											
T1PAS7,9,14	*S. sciuri*	T	post-weaning	animal				X				X
T1PAS12	*S. pasteuri*	T	post-weaning	animal				X				X
T1PHS1,2	*S. sciuri*	T	post-weaning	environment				X				X
T1FAS1,4,6,13	*S. pasteuri*	T	finishing	animal				X				X
	(MW768095)											
T1FAS5	*S. sciuri*	T	finishing	animal				X				X
	(MW768105)											
T1FAS7,15	*S. haemolyticus*	T	finishing	animal		X	X			X	X	
	(MW768097)											
T1FHS2	*S. sciuri*	T	finishing	environment				X				X
	(MW768104)											
COL	*S. aureus* (AAW37420.1)			human				X				X
MW2	*S. aureus* (WP_001801873.1)			human								
N315	*S. aureus* (BAB41256.1)			human								
SMK37o	*S. equorum*			cat	X			X	X			X
	(GU301099.1)											

Letters are used in the table to indicate amino acids (A = alanine, D = aspartic acid, E = glutamic acid, G = glycine, R = arginine, S = serine, T = threonine, Y = tyrosine) and nucleic acid bases (A = adenine, G = guanine, T = thymine).

## Data Availability

Nucleotide sequences presented in this study are openly available in GenBank (https://www.ncbi.nlm.nih.gov/genbank/, accessed on 26 May 2021).

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
