# Peer review of "Occurrence of Methicillin-Resistant Coagulase-Negative Staphylococci (MRCoNS) and Methicillin-Resistant Staphylococcus aureus (MRSA) from Pigs and Farm Environment in Northwestern Italy"

_antibiotics, 2021, doi:10.3390/antibiotics10060676_

Round 1
Reviewer 1 Report
Reviewing Manuscript number: antibiotics-1233104 entitled "Detection of methicillin-resistant coagulase-negative staphylococci (MRCoNS) and methicillin-resistant Staphylococcus aureus (MRSA) from pigs and farm environment in northwestern Italy". Regarding the manuscript antibiotics-1233104, there are some points that need clarification and some minor corrections.
Major points:
- Nucleotide sequences are not available on the GenBank under the accession number provided in the manuscript.
- Results provided as the number of positive in each farm and production type. However, proportions should also add to be able to compare between groups.
- Also, there is no statistical analysis performed to show if there is significant differences in the proportion of MRS between groups.
- Table 4 unclear and the authors already listed the mutations found in each isolate in the text, so Table 4 considers repetition.
- An ethical approval number should be provided.
Minor points:
Line 12: Swine farming as “a” source of methicillin-resistant
Line 16: Here we investigated MRSA and MRCoNS in animals … Change to “In the present study we investigated MRSA …”
Line 17: in five pigsties (four intensive, of which one antibiotic-free at finishing, and one organic) in a high 17 farm-density area of northwestern Italy. Please rephrase this sentence to reflect the type of the five farms.
Line 68-70: these two lines should be removed from the introduction as it considered as a result.
Line 99: The lowest number of 99 isolates was recovered … change to “The lowest number of 99 MRS isolates was recovered”
Line 103: Table 3 was cited in the text before Table 2!
Line 119: for Table 1, 2 and 3, Methicillin-resistant staphylococci species isolated from swine nasal swabs … change to “Methicillin-resistant staphylococci (MRS) species isolated from swine nasal swabs”.
Line 192: change uncover to investigate.
Line 194: delete “where no study was previously carried out”.
Line 198-200: Authors compare their results to the previous prevalence reported in Italy, but they did not provide the prevalence/proportion in the result section of this study.
Line 205-207: Delete “Only the antibiotic-free finishing stage of farm G showed a low proportion of positive animals (2/15). The breeding stage (sows) showed the highest MRS prevalence (48/60).” Already listed in the results section.
Line 234, 242, 259: references cited do not follow the journal format style.
Line 269: methicillin-resistant staphylococcal species change to “MRS species”
Line 336-338: Move these two lines “The sample size was calculated to detect at least one MRSA positive sample per farm, based on a minimum expected MRSA prevalence of 10% and considering a 95% confidence level.” and them at line 327 before “During each sampling ….. “
Line 342: Please add the questionnaire as a supplementary file.
Line 394: Please add the details of the positive controls.
Line 406-407: None of the provided accession numbers have information on GenBank.
Reviewer 2 Report
Methicillin-resistant Staphylococcus aureus (MRSA) is a human pathogen and widely present in pig farms. In this study, the authors selected five pig sites (four intensive and one organic) in a high farm-density area of northwestern Italy to detect the presence of MRSA and methicillin-resistant coagulase-negative staphylococci (MRCoNS) from animals and the farm environment. MALDI-TOF was used to detect the Staphylococcus species and PCR and subsequent nucleotide sequencing was used for the species confirmation. The study found the presence of MRCoNS in all five farms. However, MRSA was detected in only one pig farm. No MRSA was found from the environmental samples from any of the farm. The predominant MRCoNS species was Staphylococcus sciuri. Sequence analysis of mecA gene that encodes the methicillin resistance penicillin-binding protein (PBP2-a) and is associated with methicillin resistance found the similar point mutation in different species suggesting horizontal transfer of mecA among the species.
This study is the first to report the prevalence of MRCoNS in pig farms in northwestern Italy. While the finding is significant to assess the public health hazard from the Pig farms in that region, including more samples from different farms (especially farms that use external animal remount from different gilts’ supplier) would strengthen the findings and depicts the true picture of the prevalence of MRSA and MRCoNS in the region. Interestingly, this study is not in agreement with the previous study that was conducted in the same region (Monaco et al, 2013; PMID: 23731504, Rodríguez-López et al, 2020; PMID: 32825203) probably due to the small sample size. However, this reviewer recommends few revisions to improve the quality of the manuscript before final publication.
Comments:
- The title of the manuscript should focus on the detection of MRCoNS, rather than stating both MRSA and MRCoNS. The important findings of this study is the presence of MRCoNS. MRSA was present in only one pig farm in this study. The title should also mention prevalence instead of detection as the study did not use any new method to detect Staphylococcus species and focuses on the presence of MRSA and MRCoNS.
- Line 90: Add the total number of swine swabs from where the 127 MRS isolates were obtained.
- Line 94: Remove the word ‘just’.
- Lines 96-97: Please give the percentage of identification along all Staphylococcus species, e.g. Staphylococcus pasteuri (%), haemolyticus (%), and so on.
- Lines 96-97: The sentence ‘MRS were present in all farms’ can be incorporated in Lines 90-91.
- Line 136: It is not clear whether mecA gene was PCR amplified from all bacterial isolates and sequenced. Please mention the number of isolates sequenced in the results section.
- Lines 196-200, and 238-239: This implies the significance of adding more farms in the sampling. Please add a sentence that this study lacks the proper sampling size to accurately describe the prevalence of MRSA in pig farms.
- Lines 269-272: This paragraph is too short to be standalone. Please merge this with previous paragraph. Also, mention what may be the reason that Farm B has the most diverse Staphylococcal species.
- Table 3 should be the primary table and moved as Table 1 with some modifications. Currently the table is showing the number of MRS and the total number of samples for animals and environment at different productive stage in each farm. Individual numbers of MRSA, MRCoNS, and MRS will give the reader clear idea how many of the total isolates were obtained from each productive phase in each of the farm, how many of them are MRSA and how many of them are MRCoNS. If possible, please provide the percentage of identification along the number.
- Table 1 and 2 can be combined together. Under each column for the MRS and other bacterial species, add two other columns, one for animals and another for farm environment.
- Figure 1 does not add anything other than visual comparison of the nucleotide and amino acid sequences. Moreover, Figure 1c data also presented in Table 4. There are 37 organisms listed in Table 4, but only 26 of them shown in Figure 1c. Therefore, Figure 1 can be deleted and only present as Table. Table 4 should have the following column (From left to right) Strain, Organism, Farm, Productive phase, Sample, Point mutations, and PBP2-a mutation. The percentage of isolates that have specific mutation can be shown as separate table.
- Please check the bibliography for the correct formatting.
